# Numerical Investigation of Microporous Insulation for Power Reduction in Zero-Heat-Flux Thermometry

**DOI:** 10.3390/mi16111271

**Published:** 2025-11-12

**Authors:** Dong-Jin Lee, Dae Yu Kim

**Affiliations:** 1Center for Sensor Systems, Inha University, Incheon 22212, Republic of Korea; djlee@inha.ac.kr; 2Inha Research Institute for Aerospace Medicine, Inha University, Incheon 22212, Republic of Korea; 3Department of Electrical and Computer Engineering, College of Engineering, Inha University, Incheon 22212, Republic of Korea

**Keywords:** zero-heat-flux thermometry, microporous insulation, reduction in power consumption

## Abstract

Zero-heat-flux (ZHF) thermometry is a clinically validated method for non-invasive core body temperature monitoring, yet its broad adoption in wearable applications is constrained by the high power consumption of the heater element. In this study, we numerically investigate the role of microporous insulation in minimizing energy demand while preserving measurement accuracy. A three-dimensional finite element model of a ZHF probe was implemented in COMSOL Multiphysics 5.4, consisting of a resistive heater, a microporous insulation shell, and a skin-equivalent substrate regulated by proportional–integral–derivative (PID) control. A Taguchi L9 orthogonal array was utilized to systematically investigate the effects of porosity (0–90%), insulation thickness (2–4 mm), and the convective heat transfer coefficient (5–15 W/m^2^·K) on the thermal performance of the ZHF thermometry system. Two performance metrics—heater energy consumption and settling time—were analyzed using analysis of variance (ANOVA). The results indicated that porosity accounted for more than 95% of the variance in heater power and over 80% of the variance in settling time. The configuration with φ = 90% and t = 3 mm demonstrated a balanced trade-off between energy efficiency and transient response for low-power ZHF thermometry. These findings provide design insights for energy-efficient wearable temperature sensors.

## 1. Introduction

The accurate measurement of core body temperature is essential in clinical diagnostics, perioperative care, critical care, and continuous health monitoring [1]. Deviations in deep body temperature are early indicators of infection, hypothermia, heat stress, or metabolic imbalance, making reliable measurement a cornerstone of patient management [2,3,4,5]. Conventional invasive techniques, such as esophageal, rectal, or bladder thermometry, provide high accuracy but are unsuitable for continuous use outside intensive care environments due to their discomfort and impracticality. Non-invasive methods such as tympanic or oral thermometry are widely used, yet they are prone to environmental artifacts and are not reliable for long-term monitoring.

The zero-heat-flux (ZHF) method addresses many of these limitations by establishing a thermally neutral boundary at the skin surface [6]. In a ZHF probe, a heater and an insulation layer are applied over the skin to suppress heat flux between the body and the environment. Once equilibrium is achieved, the skin surface temperature approximates the deep tissue temperature, allowing for continuous and non-invasive monitoring. ZHF thermometry has been validated in surgical and intensive care settings and is increasingly considered for portable and wearable devices [7,8,9,10,11,12,13,14,15,16,17].

However, a persistent limitation of ZHF sensors is their high heater power consumption. The heater must continuously supply energy to compensate for losses through the insulation layer into the ambient environment. Increasing the thickness of polymeric foams can mitigate heat leakage; however, this approach leads to bulky devices that negatively affect user comfort and wearability. In battery-powered or wireless health monitoring platforms, heater consumption dominates power demand, shortening device autonomy and limiting integration with wearable electronics.

Recent advances in materials science suggest new opportunities for improving thermal management in wearable devices. Porous polymers, aerogels, and hybrid composites have been engineered to provide low thermal conductivity while maintaining mechanical strength. Among the available strategies, microstructuring with air inclusions is particularly appealing. Given that air exhibits one of the lowest thermal conductivities of common materials (~0.026 W/m·K at room temperature), incorporating air voids within an insulating polymer can significantly reduce the effective thermal conductivity [18,19]. This concept has been widely adopted in thermal insulation foams and packaging materials but has not been systematically investigated in the context of ZHF thermometry.

In parallel, non-contact and data-driven thermal control have further expanded the design space for flexible heating systems. Ganguly et al. developed a photopolymerized polypyrrole/graphene nanofiber/iron oxide nanocomposite that enabled efficient and uniform non-contact heating across flexible substrates, demonstrating how hybrid conductive networks can sustain thermal uniformity while minimizing direct heat losses—principles directly relevant to enhancing ZHF heater efficiency and user comfort [20]. Complementarily, Qi et al. highlighted the potential of machine-learning-based heat flux prediction in complex thermal systems, showing that AI-assisted modeling can capture nonlinear, multiphysics interactions with high fidelity [21].

Building on these advancements, the present study numerically investigates the role of microporous insulating layers in minimizing heater power consumption in ZHF thermometry. Using finite element analysis in COMSOL Multiphysics, we evaluate the influence of three key factors: porosity (fill factor), insulation thickness, and convective heat transfer coefficient. A Taguchi L9 orthogonal array design of experiments was implemented to efficiently explore the parameter space and identify the dominant contributors to energy demand, response time, and thermal uniformity. The outcomes provide a quantitative framework for balancing power efficiency, response dynamics, and environmental robustness, offering design guidelines for next-generation, low-power ZHF thermometers optimized for wearable healthcare.

## 2. Methods

### 2.1. Device Modeling and Geometry

A three-dimensional finite element model of a ZHF probe was constructed using COMSOL Multiphysics 5.4 (COMSOL Inc., Burlington, MA, USA). The model consisted of a resistive heater, a microporous polyethylene insulation shell, and a skin-equivalent substrate representing biological tissue. The skin was modeled as a semi-infinite solid domain (10 mm thickness) with constant thermal conductivity, density, and heat capacity corresponding to soft tissue. The heater was implemented as a volumetric heat source and controlled by a proportional–integral–derivative (PID) scheme to maintain thermal equilibrium between the heater surface and the underlying tissue.

### 2.2. Microporous Insulation Layer

The insulation layer was represented as porous polyethylene, with its effective thermal conductivity (*k_eff_*) varying as a function of porosity (φ). The values of
$k_{\mathrm{eff}}$ for polyethylene foam were determined by combining empirical data and theoretical predictions following the methodology described in [22]:
$$K_{eff}=-3.032\times 10^{-7}\times \varphi^{3}+5.307\times 10^{-5}\times \varphi^{2}-0.00542\times \varphi +0.3408.$$

### 2.3. Control Strategy (PID Implementation)

A proportional–integral–derivative (PID) control scheme was implemented within COMSOL using the Global Equations interface. The control law was expressed as
$$u\left(t\right)=K_{p}e\left(t\right)+K_{i}\int e\left(t\right)dt+K_{d}\frac{de}{dt},$$ where
$e\left(t\right)=T_{top}(t)-T_{bot}(t)$ is the instantaneous temperature error,
$T_{top}(t)$ is the heater surface temperature, and
$T_{bot}(t)$ is the skin temperature. The PID output
$u\left(t\right)\;$was linked directly to the electrode terminal voltage, and saturation limits were imposed to reflect maximum permissible heater power. Global ODEs were employed to evolve the integral and derivative states in time, ensuring real-time feedback during transient simulation.

### 2.4. Statistical Analysis

Analysis of variance (ANOVA) was conducted to quantify the contribution of each factor (φ, t, h) to variations in heater energy and settling time, using the sum of squares (SS) and percentage contribution. A significance level of *p* < 0.05 was adopted. Additionally, analysis of means (ANOM) plots were generated to visualize the influence of each parameter on performance metrics, providing guidance for optimal factor levels.

## 3. Results and Discussion

Figure 1 presents the conceptual design and numerical modeling framework of the proposed ZHF thermometry system employing a microporous insulation layer. As shown in Figure 1a, the sensing probe consists of a resistive heater embedded within a microporous polyethylene shell, with thermistors located at both the heater surface and beneath the skin interface. The addition of an outer insulation shell minimizes environmental heat leakage, enabling a thermally neutral boundary at the skin surface. This architecture is intended to improve accuracy in non-invasive core body temperature measurement while reducing heater energy demand.

The insulating performance of the porous polyethylene layer is characterized in Figure 1b. With increasing porosity, the effective thermal conductivity decreases due to the presence of low-conductivity air pockets, which suppress heat flux through the insulation. This trend highlights the potential of microporous structuring to reduce steady-state heater power while maintaining sufficient thermal isolation.

The dynamic performance under PID control is illustrated in Figure 1c. The heater temperature (*T_top_*) and interface temperature (*T*_bot_) converge toward the equilibrium condition, while the power consumption profile indicates active compensation by the feedback loop. The inset diagram depicts the PID regulation strategy, where proportional, integral, and derivative terms modulate heater input to minimize the temperature error.

Finally, Figure 1d shows the simulated temperature distribution of the ZHF sensor during operation. At the initial stage, significant gradients exist between the heater and skin due to external losses. However, as the system reaches equilibrium under PID regulation, the surface temperature aligns closely with the core body temperature (*T*_core_). These results confirm that combining microporous insulation with active thermal control enhances measurement stability and reduces power requirements in wearable ZHF thermometry.

Figure 2 illustrates the design principle, structural configuration, and thermal–electrical performance of the spiral resistive heater integrated into the ZHF thermometry probe. As shown in Figure 2a, the heater adopts an Archimedean spiral geometry, where the incremental radial distance (Δr) between adjacent traces is defined by the relation Δr = 2πb, ensuring continuous spacing along the spiral path. This configuration maximizes the effective heating area while maintaining compact dimensions, allowing localized and efficient energy delivery to the skin-contact surface. A cross-sectional schematic of the heater fabrication is presented in Figure 2b. The heater consists of copper traces deposited on an FR-4 substrate, with a thickness of 0.02 mm and trace spacing of 0.2 mm. This geometry follows established design practices for microheater structures on FR-4 substrates. Previous studies have reported copper foil thicknesses of 0.018–0.035 mm and conductor widths in the 0.1–0.2 mm range, which provide uniform heating performance, manufacturability, and mechanical stability in flexible and PCB-based microsystems [23,24,25]. This layered structure provides both electrical conductivity for Joule heating and mechanical robustness for integration with polymer-based insulation. The simulated potential distribution under a 0.2 V bias is shown in Figure 2c. The potential gradient across the spiral is smooth, confirming uniform current density along the trace and minimizing localized hot spots. Such uniform electrical characteristics are essential for stable and predictable thermal output during closed-loop control. The thermal field generated by the spiral heater is shown in Figure 2d. The heater surface exhibits a nearly symmetric temperature distribution, with the central region reaching ~39 °C while the periphery remains slightly cooler. This uniformity ensures accurate maintenance of the ZHF condition at the probe–skin interface, a prerequisite for reliable deep-body temperature estimation. Finally, Figure 2e presents the transient thermal–electrical performance of the heater. Stepwise voltage inputs (0.2–1.0 V) result in proportional increments in surface temperature, confirming controllability and repeatability of the heating process. The temperature increases rapidly upon each voltage step and stabilizes within a few minutes, demonstrating the heater’s suitability for real-time PID-controlled ZHF applications.

To systematically evaluate the influence of multiple design and environmental parameters on ZHF thermometry performance, a Taguchi design of experiments (DOE) approach was adopted. Table 1 presents the L9 orthogonal array, which was selected because it allows three control factors at three levels each to be tested efficiently with only nine simulation runs. The three factors investigated were: (1) Porosity of the microporous polyethylene layer (0%, 30%, 90%), which directly governs the effective thermal conductivity and thus influences heater power demand; (2) Insulation thickness (2, 3, 4 mm), which modifies the thermal resistance between the heater and environment; (3) Convective heat transfer coefficient (h) (5, 10, 15 W/m^2^K), which represents different ambient cooling conditions ranging from weak natural convection to stronger forced convection.

Figure 3 illustrates the transient thermal and power responses of the ZHF thermometry probe across the nine experimental runs prescribed by the Taguchi L9 orthogonal array. Each run represents a unique combination of porosity (0, 30, and 90%), insulation thickness (2, 3, and 4 mm), and convective heat transfer coefficient (5, 10, and 15 W/m^2^·K). The probe response is characterized by the evolution of the top surface temperature (T_top_), skin-side temperature (T_bot_), and heater power input (P) under proportional–integral–derivative (PID) control. The red dashed line in each subplot denotes the settling time (t_s_), defined as the moment when the temperature error (T_top_ − T_bot_) falls within the tolerance band (±0.037 °C) around the core body temperature (T_core_ = 37 °C). This tolerance was chosen to impose a more stringent numerical convergence criterion than that used in clinically validated ZHF thermometers, which typically exhibit an accuracy range of ±0.05 °C [8]. Across all cases, the heater initiates with a high power demand during the early transient period to rapidly suppress heat flux between the skin and environment. This is accompanied by a sharp rise in T_top_, followed by convergence toward T_bot_ as the thermal neutral plane is established. Quantitative analysis of this overshoot revealed values ranging from 22–30% (45–48 °C) at 0% porosity to 34–40% (50–52 °C) at 90% porosity (Table A1). The higher overshoot observed at elevated porosity levels stems from reduced effective thermal conductivity, which limits heat dissipation through the insulation layer. At steady state, the heater power demand decreased with increasing porosity—from 170–190 J at 0% porosity to 75–95 J at 90%—corresponding to nearly 50% energy savings. However, this improvement in power efficiency was accompanied by a longer settling time, extending from approximately 20–24 min at low porosity to 27–29 min at high porosity. The intermediate porosity case (30%) offered a practical compromise, achieving moderate settling time with substantially reduced energy demand. The obtained settling times were consistent with the thermal time constants reported for multilayer ZHF thermometry systems in the literature, confirming that the simulated transient behavior closely reflects the realistic performance characteristics of practical ZHF probes [26].

The Taguchi design allowed a systematic evaluation of the relative effects of each factor. Subsequent ANOVA revealed that porosity contributed most strongly to variations in power consumption (>70%), whereas convective heat transfer coefficient dominated the settling time response (>60%). The analysis of means (ANOM) further indicated that higher porosity and greater insulation thickness were beneficial for power reduction, while lower convection coefficients and moderate thicknesses accelerated transient convergence. These results highlight the inherent trade-off between energy efficiency and dynamic response in ZHF thermometry, suggesting that optimal designs must balance steady-state heater demand with clinically acceptable response times.

To quantitatively evaluate the relative influence of porosity, insulation thickness, and the convective heat transfer coefficient on ZHF thermometry performance, an analysis of variance (ANOVA) was performed for both steady-state power consumption and settling time (Table 2 and Table 3).

For power consumption, porosity emerged as the overwhelmingly dominant factor, contributing 97.6% of the total variance (*p* = 0.0118), while thickness (0.05%) and convective coefficient (1.16%) had negligible effects. This result aligns with the effective thermal conductivity analysis, where higher porosity drastically reduced heat leakage and heater demand. Thus, porosity is the principal design parameter for minimizing long-term energy consumption in wearable ZHF probes.

In contrast, the ANOVA for settling time revealed that porosity also exerted the largest influence (83.88%, *p* = 0.0036), but insulation thickness played a secondary yet significant role (15.55%, *p* = 0.019). A higher porosity substantially delayed equilibrium, as the reduced effective conductivity slowed the rate of heat transfer from the heater to the skin interface. Similarly, thicker insulation layers increased thermal resistance, further extending the transient response. The convective heat transfer coefficient was statistically insignificant (<1% contribution, *p* = 0.5309), suggesting that external convection effects are minimal compared to the intrinsic insulation properties.

Taken together, the ANOVA results highlight a fundamental design trade-off: (1) Increasing porosity reduces steady-state power consumption, enhancing energy efficiency; (2) However, the same porosity increase prolongs settling time, limiting transient responsiveness; (3) Insulation thickness provides an additional lever for tuning response dynamics but at the cost of slower equilibration when increased.

These findings underscore the necessity of balancing energy efficiency and dynamic accuracy in the optimization of microporous ZHF thermometry. A moderate porosity and optimized thickness may provide the best compromise, ensuring both low power demand and clinically acceptable response times for continuous monitoring applications.

Figure 4 presents the main effect plots derived from the Taguchi L9 orthogonal array experiments, illustrating how porosity, insulation thickness, and convective heat transfer coefficient affect energy consumption (Figure 4a–c) and settling time (Figure 4d–f).

For energy consumption, porosity emerged as the dominant parameter (Figure 4a). As porosity increased from 0% to 90%, the energy consumption decreased drastically from ~170 J to ~75 J, confirming that embedding micropores within the polyethylene insulation effectively lowers the effective thermal conductivity and suppresses heat leakage. In contrast, thickness (Figure 4b) and convection coefficient (Figure 4c) had minimal influence on steady-state power demand, as also supported by ANOVA results showing <2% contribution.

For settling time, the trend was reversed (Figure 4d). Higher porosity markedly increased the equilibration time from ~22 min at 0% porosity to ~29 min at 90% porosity. This delay arises from reduced effective conductivity, which slows heat transfer to the skin interface and prolongs the time required for the top and bottom sensors to equilibrate. Thickness also exerted a moderate influence (Figure 4e), with thicker insulation extending the settling time by adding thermal resistance. Meanwhile, convection coefficient again showed negligible effect (Figure 4f), confirming that external heat transfer plays only a secondary role compared to insulation properties.

Taken together, the main effect analysis reinforces the design trade-off identified in ANOVA: increasing porosity significantly reduces heater energy demand (enhancing energy efficiency) but at the expense of longer settling times (slower dynamic response). Thickness provides a secondary tuning parameter for transient performance, while the role of convective heat transfer coefficient is negligible under the studied conditions.

## 4. Conclusions

This study numerically investigated the use of microporous polyethylene insulation in ZHF thermometry, with a focus on reducing heater energy consumption while maintaining reliable dynamic performance. A Taguchi L9 orthogonal array was employed to systematically assess the effects of porosity, insulation thickness, and the convective heat transfer coefficient on probe performance.

The results demonstrated that porosity is the dominant factor influencing both steady-state energy consumption and thermal settling time. Increasing porosity drastically reduced effective thermal conductivity, leading to up to ~55% lower heater energy demand compared with non-porous insulation. However, this improvement in energy efficiency was accompanied by a longer settling time, reflecting a trade-off between power savings and response speed. Thickness exhibited a secondary effect, particularly on settling time, while the convective heat transfer coefficient showed only marginal influence within the tested range.

Analysis of variance (ANOVA) confirmed that porosity accounted for more than 95% of the variation in heater power and over 80% of the variation in settling time, highlighting its critical role in design optimization. The configuration with φ = 90% and t = 3 mm represents a balanced trade-off between energy efficiency and transient response for wearable, low-power ZHF thermometry. Nevertheless, this configuration should not be regarded as universally optimal, as the preferred design is application-dependent—wearable sensors prioritize reduced power consumption for prolonged monitoring, whereas clinical probes may favor faster thermal equilibration even at the expense of higher power usage.

Although repeated heating–cooling cycles were not explicitly simulated, the stable temperature convergence within ±0.037 °C under PID control indicates strong intrinsic thermal stability of the proposed ZHF system. These findings establish microporous insulation as an effective strategy for achieving low-power, wearable ZHF thermometry, enabling extended operational lifetime in battery-driven and wireless health monitoring platforms. Future studies will focus on experimental validation, cyclic stability evaluation, and integration with flexible electronic architectures to translate these design insights into practical biomedical applications.

## Figures and Tables

**Figure 1 micromachines-16-01271-f001:**
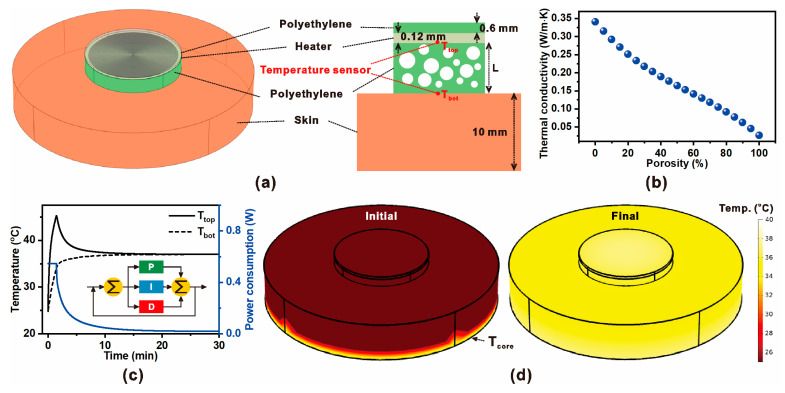
(**a**) Schematic and modeling concept of ZHF thermometry incorporating a microporous polyethylene insulation layer, resistive heater, temperature sensors placed above and below the skin, and an external insulation shell. (**b**) Effective thermal conductivity of the polyethylene insulation as a function of porosity, demonstrating the reduction in thermal conductivity with increasing micropore fraction. (**c**) Transient temperature responses of the heater surface (T_top_) and interface (T_bot_) along with power consumption profile under proportional–integral–derivative (PID) control; inset shows the closed-loop regulation scheme. (**d**) Simulated temperature distribution of the ZHF probe at the initial stage of thermal imbalance and at the final equilibrium state, where the measured skin-surface temperature converges toward the core body temperature (T_core_).

**Figure 2 micromachines-16-01271-f002:**
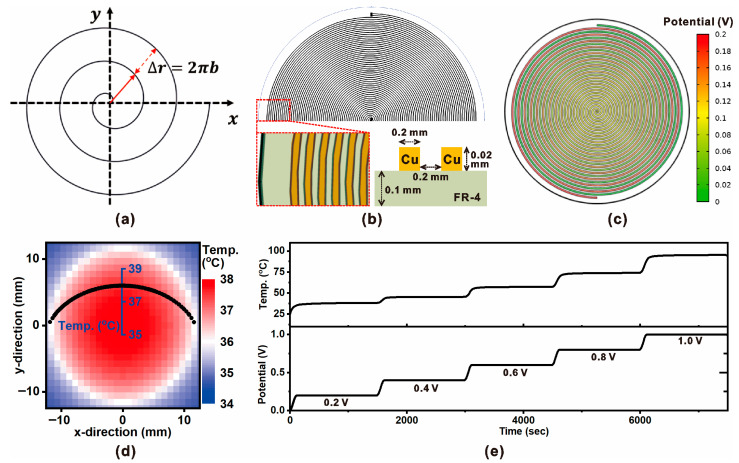
Design and electrothermal simulation of an Archimedean spiral microheater for zero-heat-flux applications. (**a**) Schematic of the spiral electrode geometry, where the radial increment is given by Δr = 2πb, with b as the spiral pitch constant. (**b**) Cross-sectional schematic of the fabricated copper–FR4 heater structure, showing Cu trace thickness and spacing. (**c**) Simulated potential distribution across the spiral heater pattern under an applied voltage of 0.2 V, demonstrating uniform current flow. (**d**) Steady-state temperature distribution at the heater surface, indicating a uniform heating profile across the active area. (**e**) Transient electrical and thermal responses of the spiral heater: applied potential steps (0.2–1.0 V) result in proportional increases in surface temperature, confirming controllable heating performance.

**Figure 3 micromachines-16-01271-f003:**
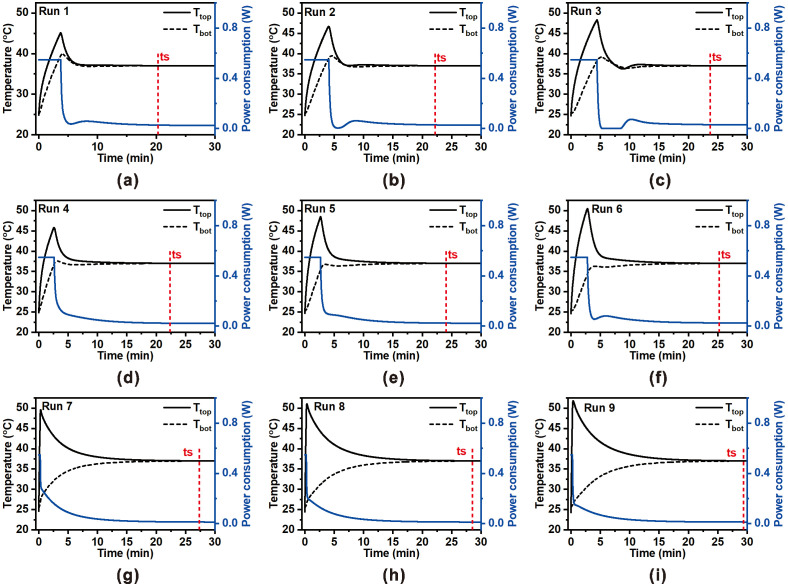
Transient thermal and power responses of the ZHF thermometry probe for the nine Taguchi L9 orthogonal array runs. Each subplot (**a**–**i**) presents the top surface temperature (T_top_, solid black), skin-side temperature (T_bot_, dashed black), and heater power consumption (P, blue). The red dashed line indicates the settling time (t_s_), defined as the moment when the temperature error (T_top_ − T_bot_) falls within the tolerance band of the core temperature (T_core_). The nine cases correspond to different combinations of porosity (0, 30, 90%), insulation thickness (2, 3, 4 mm), and convective heat transfer coefficient (5, 10, 15 W/m^2^·K), arranged according to the Taguchi orthogonal array.

**Figure 4 micromachines-16-01271-f004:**
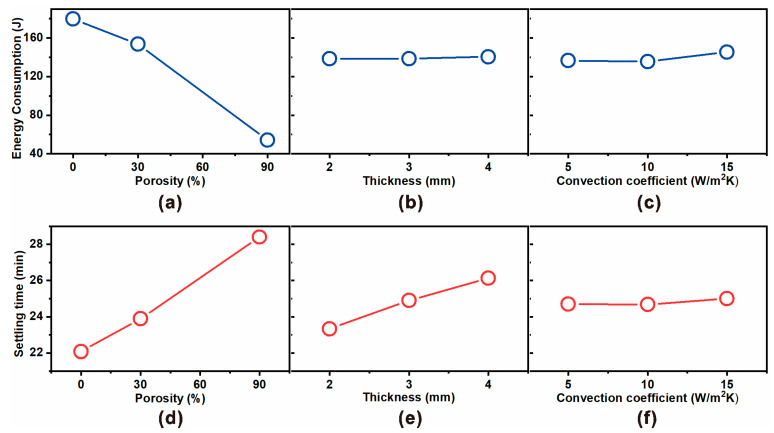
Main effect plots for (**a**–**c**) energy consumption and (**d**–**f**) settling time as functions of porosity, insulation thickness, and convective heat transfer coefficient, based on Taguchi L9 orthogonal array analysis. Porosity exhibits the strongest effect, where increasing porosity significantly decreases heater energy demand but simultaneously prolongs the thermal settling time. Thickness moderately influences the settling time while having negligible impact on power, whereas convection coefficient shows only minor effects on both responses.

**Table 1 micromachines-16-01271-t001:** Taguchi L9 orthogonal array experimental design with three control factors—porosity of the microporous insulation layer, insulation thickness, and convective heat transfer coefficient (*h*).

Run	Porosity (%)	Thickness (mm)	Convective Heat Transfer Coefficient (W/m^2^K)
1	0	2	5
2	0	3	10
3	0	4	15
4	30	2	10
5	30	3	15
6	30	4	5
7	90	2	15
8	90	3	5
9	90	4	10

**Table 2 micromachines-16-01271-t002:** Analysis of variance (ANOVA) results for heater power consumption in ZHF thermometry as a function of porosity, insulation thickness, and convective heat transfer coefficient.

Factor	Degree of Freedom (DOF)	Sum of Squares (SS) (J^2^)	*p*-Value (*p*)	Percentage Contribution
Porosity	2	14639.24	0.0118	97.6%
Thickness	2	7.62	0.9582	0.05%
h coefficient	2	174.56	0.5001	1.16%
Residual	2	174.60		1.16%

**Table 3 micromachines-16-01271-t003:** Analysis of variance (ANOVA) results for settling time in ZHF thermometry as a function of porosity, insulation thickness, and convective heat transfer coefficient.

Factor	Degree of Freedom (DOF)	Sum of Squares (SS) (min^2^)	*p*-Value (*p*)	Percentage Contribution
Porosity	2	63.72	0.0036	83.88%
Thickness	2	11.82	0.019	15.55%
h coefficient	2	0.2	0.5309	0.27%
Residual	2	0.23		0.30%

## Data Availability

The original contributions presented in this study are included in the article. Further inquiries can be directed to the corresponding author.

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
