# Peer review of "Numerical Investigation of Microporous Insulation for Power Reduction in Zero-Heat-Flux Thermometry"

_micromachines, 2025, doi:10.3390/mi16111271_

Round 1

Reviewer 1 Report

Comments and Suggestions for Authors

The manuscript presents a  numerical study on reducing power consumption in Zero-Heat-Flux (ZHF) thermometry, a critical challenge for its adoption in wearable devices. The application of microporous insulation is a well-motivated approach, and the use of Finite Element Analysis (FEA) combined with the Taguchi L9 orthogonal array for systematic parameter screening is methodologically sound and efficient. The manuscript is generally well-written and structured. However, several key aspects require clarification and strengthening before the manuscript can be considered for publication.

  1. The settling time (tₛ) is defined as the point where the temperature error falls within the "tolerance band of the core body temperature." The specific value of this tolerance band (e.g., ±0.1°C) is a critical parameter that is not stated. This value must be provided, as it directly influences the reported settling times.
  2. The trade-off between power consumption and response speed has been clearly illustrated. However, the conclusion suggesting a configuration of 90% porosity and 3 mm thickness as the "optimal configuration" requires further elaboration. The discussion should emphasize that the "optimum" is highly application-dependent and should specify the particular application requirements involved.

Author Response

Reviewer 1, General comment: The manuscript presents a numerical study on reducing power consumption in Zero-Heat-Flux (ZHF) thermometry, a critical challenge for its adoption in wearable devices. The application of microporous insulation is a well-motivated approach, and the use of Finite Element Analysis (FEA) combined with the Taguchi L9 orthogonal array for systematic parameter screening is methodologically sound and efficient. The manuscript is generally well-written and structured. However, several key aspects require clarification and strengthening before the manuscript can be considered for publication.

Author response: We sincerely thank the reviewer for the encouraging and constructive summary. We appreciate the recognition of the novelty and methodological rigor of our work. In response to the reviewer’s request for clarification and strengthening, we have carefully revised the manuscript to improve both technical detail and presentation clarity throughout.

Reviewer 1, Comment #1: The settling time (tₛ) is defined as the point where the temperature error falls within the "tolerance band of the core body temperature." The specific value of this tolerance band (e.g., ±0.1°C) is a critical parameter that is not stated. This value must be provided, as it directly influences the reported settling times.

Author response: We appreciate the reviewer’s insightful comment. Clinically validated zero-heat-flux (ZHF) thermometers typically exhibit an accuracy of ±0.05 °C [R1]. To impose a more stringent numerical convergence criterion in our simulation, we defined a tolerance band of ±0.037 °C, corresponding to 0.1 % of the core body temperature (37 °C). This criterion ensures that the temperature difference between the heater surface (Ttop) and the skin-contact interface (Tbot) is effectively negligible, thereby representing a true steady-state condition of ZHF thermometry.

R1: Wang, J.Y.; Liang, H.; Tian, C.Z.; Rong, G.Y.; Shao, X.F.; Ran, C. Agreement of zero-heat-flux thermometry compared with infrared tympanic temperature monitoring in adults undergoing major surgery. European Journal of Medical Research 2025, 30, doi:10.1186/s40001-025-02317-9.

Author action: We amended the manuscript as following:

  • Page 6, Line 211, “The red dashed line in each subplot denotes the settling time (ts), defined as the moment when the temperature error (Ttop – Tbot) falls within the tolerance band (±0.037 °C) around the core body temperature (Tcore=37 °C). This tolerance was chosen to impose a more stringent numerical convergence criterion than that used in clinically validated ZHF thermometers, which typically exhibit an accuracy range of ±0.05 °C [8].”

Reviewer 1, Comment #2: The trade-off between power consumption and response speed has been clearly illustrated. However, the conclusion suggesting a configuration of 90% porosity and 3 mm thickness as the "optimal configuration" requires further elaboration. The discussion should emphasize that the "optimum" is highly application-dependent and should specify the particular application requirements involved.

Author response: We appreciate the reviewer’s thoughtful comment. We agree that the “optimal configuration” should be interpreted within the context of specific application requirements. The configuration with 90 % porosity and 3 mm insulation thickness is described as a balanced design, effectively minimizing steady-state power consumption while maintaining acceptable response speed under wearable, battery-powered operation, where long-term energy efficiency is prioritized. In contrast, for clinical or perioperative applications that demand rapid temperature stabilization, a configuration with lower porosity or thinner insulation may be more appropriate, even at the expense of increased power consumption.

Author action: We amended the manuscript as following:

  • Conclusions, Page 10, “The configuration with φ = 90 % and t = 3 mm represents a balanced trade-off between energy efficiency and transient response for wearable, low-power ZHF thermometry. Nevertheless, this configuration should not be regarded as universally optimal, as the preferred design is application-dependent—wearable sensors prioritize reduced power consumption for prolonged monitoring, whereas clinical probes may favor faster thermal equilibration even at the expense of higher power usage.”

Reviewer 2 Report

Comments and Suggestions for Authors

The manuscript entitled, ‘Numerical Investigation of Microporous Insulation for Power Reduction in Zero-Heat-Flux Thermometry’ reported Investigation of Microporous Insulation. The article should be modified according the following comments:

  1. The abstract lacks specificity regarding the data presented in the study. It is recommended to highlight key findings or notable data to give readers a clearer understanding of the study's contributions.
  2. What was the rationale behind selecting 0.02 mm thickness and 0.2 mm spacing? Were these parameters experimentally optimized?
  3. How stable is the thermal output over time during repeated heating–cooling cycles?
  4. Were any transient thermal response times (e.g., time to reach steady-state temperature) quantified and compared to design expectations?
  5. What is the power efficiency of the heater under the 0.2–1.0 V input range, and how does this influence energy consumption in practical operation?
  6. The text mentions a “sharp rise in Ttop” and variable overshoot. Can the authors provide quantitative data (e.g., overshoot percentage, peak temperature) to compare across conditions?
  7. Some articles would be significance for your reference:
  • Ganguly, S., Kanovsky, N., Das, P., Gedanken, A., & Margel, S. (2021). Photopolymerized thin coating of polypyrrole/graphene nanofiber/iron oxide onto nonpolar plastic for flexible electromagnetic radiation shielding, strain sensing, and non‐contact heating applications. Advanced materials interfaces8(23), 2101255.
  • Qi, S., Han, B., Zhu, X., Yang, B. W., Xing, T., Liu, A., & Liu, S. (2025). Machine learning in critical heat flux studies in nuclear systems: A detailed review. Progress in Nuclear Energy179, 105535.

Author Response

The file is attached.

Round 2

Reviewer 2 Report

Comments and Suggestions for Authors

This can be published in its present form.